# In the Absence of a TCR Signal IL-2/IL-12/18-Stimulated γδ T Cells Demonstrate Potent Anti-Tumoral Function Through Direct Killing and Senescence Induction in Cancer Cells

**DOI:** 10.3390/cancers12010130

**Published:** 2020-01-04

**Authors:** Karin Schilbach, Christian Welker, Naomi Krickeberg, Carlotta Kaißer, Sabine Schleicher, Hisayoshi Hashimoto

**Affiliations:** Department of Pediatric Hematology and Oncology, University Children’s Hospital Tuebingen, Hoppe-Seyler Street 1, 72076 Tübingen, Germany; c_welker@web.de (C.W.); naomi@krickeberg.de (N.K.); carlotta.kaisser@student.uni-tuebingen.de (C.K.); Sabine.Schleicher@med.uni-tuebingen.de (S.S.); hisayoshi.hashimoto@med.uni-tuebingen.de (H.H.)

**Keywords:** γδ T cells, IL-12, IL-18, senescence, TCR bypass stimulation

## Abstract

Abundant IFN-γ secretion, potent cytotoxicity, and major histocompatibility complex-independent targeting of a large spectrum of tumors make γδ T cells attractive candidates for cancer immunotherapy. Upon tumor recognition through the T-cell receptor (TCR), NK-receptors, or NKG2D, γδ T cells generate the pro-inflammatory cytokines TNF-α and IFN-γ, or granzymes and perforin that mediate cellular apoptosis. Despite these favorable potentials, most clinical trials testing the adoptive transfer of pharmacologically TCR-targeted and expanded γδ T cells resulted in a limited response. Recently, the TCR-independent activation of γδ T cells was identified. However, the modulation of γδ T cell’s effector functions solely by cytokines remains to be elucidated. In the present study, we systematically analyzed the impact of IL-2, IL-12, and IL-18 in parallel with TCR stimulation on proliferation, cytokine production, and anti-tumor activity of γδ T cells. Our results demonstrate that IL-12 and IL-18, when combined, constitute the most potent stimulus to enhance anti-tumor activity and induce proliferation and IFN-γ production by γδ T cells in the absence of TCR signaling. Intriguingly, stimulation with IL-12 and IL-18 without TCR stimulus induces a comparable degree of anti-tumor activity in γδ T cells to TCR crosslinking by killing tumor cells and driving cancer cells into senescence. These findings approve the use of IL-12/IL-18-stimulated γδ T cells for adoptive cell therapy to boost anti-tumor activity by γδ T cells.

## 1. Introduction

γδ T cells constitute a major group of T cells categorized as innate that respond rapidly to antigens and manifest immediate effector functions. Known specific stimulators of Vδ2^+^ T cells comprise the non peptidic substances such as phosphoantigens through recognition of butyrophilin-3 (BTN3), aminobisphosphonates and alkylamines, and candidate proteins involved in their activation include apolipoprotein A-I, F1-ATPase, ULBP proteins, ESAT-6, and heat shock proteins like HSP60 and HSP70 [1]. Vδ1^+^ γδ T cells recognize major histocompatibility complex (MHC) class I chain-related protein A and B (MICA and MICB), as well as foreign/self-lipid presented by non-classical MHC molecules like CD1 family [2]. Since they are activated during the early phases of infection or inflammation before effective adaptive immune responses develop and are transmitters of critical signals to activate the adaptive immune system, they act as a bridge between the innate and adaptive immune systems. The unique features of γδ T cells, such as their capability to directly recognize antigens, efficiently produce cytokines and lyse tumor cells, as well as their ability to effectively orchestrate the network of effector cells, related to their key roles in immune surveillance and protective immunity, make γδ T cells ideal candidates to establish long-lasting immunity against pathogenic invaders and cancers. Accordingly, efforts to harness γδ T cells for clinical contexts, mostly by the adoptive transfer of ex vivo activated and expanded γδ T cells, have been tested for various types of cancers, but most trials have resulted in limited clinical efficacy [3,4,5,6,7,8,9,10,11]. 

In contrast to the kingdom of adaptive immunity where αβ T cells exclusively recognize peptide epitopes presented by MHC or CD1 molecules, the nature of molecules that (may) serve as (co) ligands for the Vδ2 TCR are only gradually deciphered. Recently, TCR signals have been demonstrated to be involved in the acquisition of a regulatory phenotype in human peripheral Vδ2^+^ T cells under certain conditions [12,13,14,15]. We therefore thought to dissect the contribution of a TCR signal from microenvironmental cues such as IL-12 and IL-18. Strikingly, physiologically, both cytokines derive from myeloid cells thus innate immune cells. While TCR or NK-receptor stimulation in the presence of IL-12 and IL-18 promotes effector functions such as cytotoxicity and IFN-γ expression, and enhances the clonal expansion of Vγ9Vδ2 T cells [16,17,18,19], a synergistic effect for IL-12 and IL-18 in a TCR-independent production of IFN-γ has been demonstrated for γδ T cells lately [20,21,22]. In an effort to dissect potentially detrimental suppressive activity induced via supraphysiologic pharmacological TCR triggering from anti-tumoral behavior, we hypothesized that this capacity of γδ T cells for TCR-independent activation, namely bypassing the TCR via cytokines, might be an option to generate physiologically activated Vγ9Vδ2 γδ T cells in reasonable numbers for anti-tumoral immunotherapy [20]. While this ability of TCR-independent activation is seen to varying degrees also in conventional CD4^+^ or CD8^+^ αβ T cells, γδ T cells have the potential of secreting cytokines more abundantly than these cells [21,23]. In this context the importance of γδ T cells as an early source of IFN-γ in the treatment of cancers has been reported [24,25,26], and IFN-γ is considered a central cytokine in anti-tumor immune responses by γδ T cells [10].

In order to elucidate whether TCR-bypass-stimulated γδ T cells can be a therapeutic tool for cancers, we systematically analyzed the effects of IL-2, IL-12, IL-18 and that of TCR stimulation on γδ T cells and directly compared their impact on proliferation, activation, cytokine production, and anti-tumoral activity in vitro.

## 2. Methods

### 2.1. Bioethics

All procedures performed in this study were in accordance with the ethical standards and recommendations of the institutional ethic committee of the Medical Faculty of the Eberhard-Karls-University and the University Hospital of Tübingen and in accordance with the 1964 Helsinki declaration and its later amendments. The protocol was approved by the ethics committee under the following reference: 105/2017BO2. All subjects included in the study gave written informed consent.

### 2.2. PBMC Isolation and Culture

Peripheral blood mononuclear cells (PBMCs) from healthy donors were isolated by Ficoll density gradient centrifugation (Biocoll Separating Solution; Biochrom, Berlin, Germany). Written informed consent was obtained from each blood donor as described above. PBMCs and γδ T cells were cultured in RPMI 1640 supplemented with 10% FBS, 1% penicillin/streptomycin (all from Biochrom). IL-2 (20 or 50 IU/mL), IL-12, or IL-18 (10 ng/mL, respectively), or in combination, were added in designated experiments. 

### 2.3. Isolation of γδ T Cells from PBMCs

According to the manufacturer’s protocol, for positive isolation of γδ T cells; FITC-anti-TCRγδ antibody and anti-FITC Microbeads (Miltenyi Biotec, Bergisch Gladbach, Germany), for untouched isolation of γδ T cells; TCRγ/δ^+^ T cell isolation kit (Miltenyi Biotec), were used. The purity of γδ T cells was ≥ 97%.

### 2.4. Cell Culture of γδ T Cells

γδ T cells were negatively or positively isolated and incubated overnight at 5 × 10^5^ cells/well with 20 IU/mL IL-2 in a 96-well plate. As preparation for γδ TCR stimulation, culture plates were coated by overnight incubation with a TCRγδ antibody, IMMU510 (BD Pharmingen, San Jose, CA 95131, USA, 1 μg/mL) at 4 °C. Then the cells were seeded at 5 × 10^5^ cells/well into unmanipulated 96-well plates or those pre-coated with IMMU510, and cultured for 4 days. In addition, IL-12 and or IL-18 (10 ng/mL, respectively), IL-2 (50 U/mL) or combinations of them were added.

### 2.5. Flow Cytometric Analysis

PE-IL-4, PE-IL-17A, APC-IFN-γ, PE-granzyme B, APC-perforin (all from BioLegend, San Diego, CA 92121, USA), PerCP-anti-CD3, PerCP-anti-CD4, APC-H7-CD8, PE-TCRαβ, APC-TNF-α, FITC/PE -TCRγδ and PE-TCRVδ2 (all from BD Pharmingen), APC-NKG2D (R&D, Minneapolis, MN 55413, USA), FITC-TCRVγ9 and PE-TCRVδ2 (both from Thermo Scientific; Darmstadt, Germany), APC-FasL (Miltenyi Biotec), and FITC-Fas (Immunotech, Praha, Czech Republic) were used according to the manufacturer’s instructions. For intracellular staining of cells, the FIX & PERM^®^ Cell Permeabilization Kit (ADG Bio Research, Diepoldsau, Switzerland) was used. Cells were cultured at 37 °C for 1 h in 96-well plates in 200 μL culture medium in the presence of 1 μg Brefeldin A before staining. Samples were measured using a BD FACS CantoII flow cytometer and analyzed using Flow Jo Software.

### 2.6. Cell Trace Violet (CTV) Proliferation Assay

iIIsolated γδ T cells were labelled with CellTraceTM Violet Cell Proliferation Kit (Thermo Fisher Scientific) according to the manufacturer’s protocol and stimulated with cytokines and or the IMMU510 antibody. On day4 proliferation of γδ T cells was measured by CTV dye solution using a BD CantoII flow cytometer and analyzed using Flow Jo Software. CTV^low^ cells were calculated as proliferating cells.

### 2.7. Cytotoxicity Assay via Electric Cell-Substrates Impedance Sensing

The xCELLigence RTCA MP instrument (ACEA Biosciences, San Diego, CA, USA) was utilized for cytotoxicity assay. First, 50 μL of culture medium was added to each well of a 96 well E-plate (ACEA Biosciences) and the background impedance was measured. Target cells were seeded at a density of 1 × 10^5^ (A673) or 5 × 10^3^ (RH30 and SY5Y) cells/well in a volume of 100 μL and allowed to passively adhere on the electrode surface. Post seeding, the E-Plate was transferred to the RTCA MP instrument inside a cell culture incubator and incubated for the first 24 h without effector cells. Then, γδ T cells untreated or treated with the cytokines and/or IMMU510 were applied onto the grown tumor cell layer, with different effector cell: target cell ratios (E:T ratios; 5:1, 2.5:1, 1:1, 0.5:1) in a volume of 100 μL. Changes in impedance, which indicate the attachment and adherence of cells to the plate’s electrode, were reported every 24 h for the following 96 or 120 h. Data analysis was performed using the RTCA Software v1.2.1 (OLS).

### 2.8. Cell Cycle Analysis

T24 (bladder carcinoma), MCF7 (breast cancer) and Wm115 (melanoma) cells were plated on a 48-well plate at the concentration of 3 × 10^3^, 1.5 × 10^4^, 9 × 10^3^ cells/well, respectively and on the following day 1 × 10^4^ cells of untouched or TCR-bypass-stimulated or IMMU510/IL-12/IL-18-stimulated γδ T cells were added onto the tumor cells. After three days of co-culturing, EdU-labeling was done for 16 h using a Click-iT EdU Alexa-Fluor 647 Flow Cytometry Kit (ThermoFisher Scientific). Then, cells were harvested and analyzed for cell cycle by detecting incorporated EdU together with staining propidium iodide (PI) by using an FxCycle™ PI/RNase Staining Solution (ThermoFisher Scientific). Where indicated, neutralizing antibodies for IFN-γ and/or TNF-α were included into the culture medium to show these cytokines’ contribution to senescence induction.To neutralize IFN-γ or TNF-α, we used anti-IFN-γ antibody (clone#25718, R&D systems) at a concentration of 50 ng/mL or anti-TNF-α antibody (clone D1B4, Cell Signaling Technology, Danvers, MA, USA) at a concentration of 10 ng/mL or corresponding isotype controls. 

### 2.9. SA-β-Galactosidase Staining

MCF7 cells (1.0 × 10^4^/well) were seeded to 8-well chambered cell culture slide (ThermoFisher Scientific) for overnight before being co-cultured with untouched or TCR-bypass-stimulated or IMMU510/IL-12/IL-18-stimulated γδ T cells (1.0 × 10^4^/well). After 4 days of incubation, SA-*β*-galactosidase staining was carried out using the senescence histochemical staining kit (US Biological, Salem, MA, USA) following the manufacturer’s instruction. Cells were incubated at 37 °C for 16 h and pictures were taken.

### 2.10. RNA Extraction, cDNA Synthesis

For transcript quantification, RNA was extracted with the use of RNeasy Mini Kit (Qiagen, Hilden, Germany) and reverse transcription was carried out using Superscript III First Strand Synthesis Super Mix (Life Technology, Carlsbad, CA, USA) according to the manufacturer’s protocol.

### 2.11. Real Time PCR

For the quantitative analysis of cytokine and transcription factor expression in γδ T cells or of senescence markers in tumor cells, real-time PCR was conducted with SYBR Green kit (Promega, Walldorf, Germany) in a BioRad C1000 Thermal cycler/CFX96 real-time System (BioRad, hercules, CA, USA). Briefly, 5 ng cDNA was added to a final volume of 10 μL/reaction containing 1 × SYBR Green PCR Master Mix (Promega, Walldorf, Germany) and 100 nM of each primer. Thermal cycling conditions were: denaturation at 95 °C 10 min, 40 cycles: 95 °C/30 s, 59 °C/30 s and 72 °C, 1 min for elongation. Primers: GAPDH: forward; 5′-CCACATCGCTCAGACACCAT-3′ and reverse; 5′-GGCAACAATATCCACTTTACCAGACT-3′. GranzymeB: forward; 5′-TTCGTGCTGACAGCTGCTCACT-3′ and reverse; 5′-CTCTCCAGCTGCAGTAGCATGA-3′. Perforin: 5′-ACCAGCAATGTGCATGTGTCTG-3′ and reverse; 5′-GCCCTCTTGAAGTCAGGGT-3′, T-bet: forward; 5′-GCCTACCAGAATGCCGAGATTA-3′ and reverse; 5′-ACTCAAAGTTCTCCCGGAATCC-3′, Eomes: forward; 5′-GGCAAAGCGGACAATAACAT-3′ and reverse; 5′-AGCCTCGGTTGGTATTTGTG-3′, CDKN1A (*p*21): forward; 5′-AGGTGGACCTGGAGACTCTCAG-3′ and reverse; 5′-TCCTCTTGGAGAAGATCAGCCG-3′; CDKN2A (*p*16): forward; 5′-CTCGTGCTGATGCTACTGAGGA-3′ and reverse; 5′-GGTCGGCGCAGTTGGGCTCC-3′, *p*53: forward; 5′-CCCAACAACACCAGCTCCT-3′ and reverse; 5′-CTGGGCATCCTTGAGTTCC-3′.

### 2.12. Statistical Analysis

Statistical tests were performed with GraphPad PRISM. Comparisons between two groups for normally distributed data with equal variance were done using Student’s *t* test. For comparisons between multiple groups, one-way ANOVA followed by Tukey’s multiple comparison test was used to evaluate the statistical significance, which was considered at *p* < 0.05.

## 3. Results

### 3.1. IL-12 Combined with IL-18 Induces the Proliferation of γδ T Cells both in the Presence and Absence of TCR Stimulation

To determine the individual and synergistic effect of IL-2, IL-12 and IL-18 on the proliferation of γδ T cells, untouched isolated CFSE-labelled γδ T cells were treated with TCR stimulus through the pan-γδ antibody IMMU510 and or the cytokines, IL-2, IL-12, IL-18, or combinations thereof. Then, these cells were examined for their proliferation by flow cytometry. Both, in the presence and absence of TCR stimulus, IL-2/IL-12/IL-18 combination significantly induced the proliferation of γδ T cells compared to medium control. As shown previously [27], the anti-γδ antibody markedly increased the proliferation of γδ T cells (Figure 1).

### 3.2. γδ T Cells Produce IFN-γ, TNF-α, and IL-17 in Response to the Combination of IL-2, IL-12 and IL-18

It is known that γδ T cells exert anti-tumor activity by generating various cytokines, such as IFN-γ and TNF-α [28,29]. However, the impact of cytokines on the cytokine production of γδ T cells, especially in the absence of TCR triggering, is not well established. Therefore, in this study, γδ T cells were examined by intracellular FACS staining for their production of IFN-γ, IL-17, IL-4 and TNF-α after cytokine stimulation with or without concurrent TCR stimulation. 

By comparing stimulation with and without IMMU510, the frequency of IFN-γ-producing cells was significantly increased by TCR stimulation in context with IL-2. The addition of IL-12 and IL-18 massively increased IFN-γ-producing cells—up to 200-fold compared to control (no cytokine treatment, no TCR stimulus) and was 14-fold when simultaneously stimulated via IMMU510 compared to TCR stimulation alone-, which far exceeded the level induced by single IL-12 or IL-18 stimulation both in the absence and presence of TCR stimulus (Figure 2A). 

TNF-α production by γδ T cells seemed dependent on a combination of IL-2/IL-12 or IL-2/IL-12/IL-18. TNF-α was expressed by a slight proportion of γδ T cells (up to 5%) compared to IFN-γ and was remarkably induced in some donors with high inter-individual variances. In the presence of TCR stimulus, the combination of IL-2, IL-12 and IL-18 induced significant TNF-α production, which increased to about 30-fold of control (no cytokine treatment, no TCR stimulation) (Figure 2B). 

The frequency of IL-17-producing cells was significantly increased by TCR stimulation in context with IL-2/IL-12. In the absence of TCR stimulus, combinational treatment with IL-2/IL-12/IL-18 increased IL-17-producing cells to about 10-fold of no cytokine treatment to an absolute share of 0.8%. In the presence of a TCR stimulus, IL-2/IL-12 treatment significantly induced IL-17-producing cells to about 7-fold of no cytokine treatment with a share of 2.6% of total γδ T cells (Figure 2C).

TCR stimulation significantly elevated the frequency of IL-4-producing γδ T cells in the context of no additional cytokines, or IL-2 alone. However, no significant difference was observed in the IL-4 production induced by the individual cytokine treatments, either with or without simultaneous TCR stimulus. The overall production of IL-4 was about one-tenth of IFN-γ production (Figure 2D). Thus IL-4 production seems an intrinsic feature of γδ T cells.

As shown in Figure 2E, IFN-γ-producing cells stimulated with IL-2/IL-12/IL-18 with or without TCR stimulation did not simultaneously produce IL-17A or IL-4. On the other hand, γδ T cells producing both IFN-γ and TNF-α were detected after TCR stimulus in the presence of IL-2/IL-12/IL-18 (Figure 2F). These data clearly show a Th1 cytokine profile for IL-2/IL-12/IL-18-exposed γδ T cells. 

### 3.3. Neither T-Bet nor Eomes are Induced by Cytokine Stimulation

The transcription factor T-bet and Eomes is critical for IFN-γ expression in the mouse [30,31] and has a partially contributive role on human γδ T cells [20]. To investigate whether the cytokine-induced production of IFN-γ is associated with increased T-bet or Eomes expression, we examined the expression of T-bet and Eomes in cytokine-treated γδ T cells by qPCR. On the contrary to our expectation, compared to untreated control, the expression of T-bet was reduced to a similar extent in all cytokine treatments regardless of the presence of TCR stimulus. Neither the tested cytokines nor the TCR stimulus made significant differences (Figure 3A). The expression of Eomes was significantly downregulated by IL-12 or IL-18 or the combination of IL-12/IL-18 in the absence of TCR stimulus. TCR stimulus significantly decreased the expression of Eomes in context with IL-2/IL-12 and IL-2/IL-12/IL-18 (Figure 3B). Transcriptome analysis suggests that the enhanced production of IFN-γ by cytokines is not mediated by T-bet or Eomes, thus potentially via IκBζ [20,32,33]. However, as protein expression may differ from mRNA expression, this point needs further investigation. 

### 3.4. Granzyme B/Perforin Production by γδ T Cells is Increased by TCR Stimulation and the Combination of IL-2/IL-12/IL-18

To examine the impact of cytokine- versus TCR-mediated stimulation on the expression of cytotoxic granules involved in tumor cell killing, we analyzed the expression level of granzyme B and perforin in differently stimulated γδ T cells by qPCR. As shown in Figure 3C, TCR stimulation significant increased granzyme B production and even increased further by the addition of IL-12/IL-18. Conversely, in the absence of TCR stimulation, the expression of granzyme B was marginal in all individual cytokine settings. Unexpectedly and in contrast, perforin mRNA expression was down-regulated by TCR stimulus and further diminished by each cytokine treatment (Figure 3D). However when we investigated the protein expression of granzyme B and perforin in untouched, IL-2/IL-12/IL-18-stimulated, and IMMU510-stimulated γδ T cells by flow cytometry, the expression of granzyme B and perforin was upregulated by IL-2/IL-12/IL-18 or TCR stimulation (Figure 3E). In addition, the expression of FasL, another key mediator of apoptosis induction, was upregulated by bypass cytokine stimulation (Figure 3F). These findings suggest that γδ T cells stimulated by IL-2/IL-12/IL-18 can potentially kill tumor cells through cytotoxins and Fas-FasL interaction.

### 3.5. Expression of NKG2D on γδ T Cells is Reduced by TCR Stimulation

The NK receptor NKG2D expressed on γδ T cells contributes to their tumor cell recognition as a major killer receptor and serves as a co-stimulatory signal for cytokine production and proliferation [30,34,35,36,37]. We examined the effect of cytokines and TCR stimulus on the cell surface expression of NKG2D. Contrary to expectation, antibody-mediated TCR-triggering seemed counterproductive for the expression of NKG2D on γδ T cells in comparison to when γδ T cells were treated with cytokine(s) only. Consistently, the frequency of NKG2D expression was significantly enhanced by single cytokine regimen, i.e., IL-2, IL-12, IL-18 only (Figure 4A), and NKG2D mean fluorescence intensity (MFI) was significantly further elevated by the combination of IL-2, IL-12, and IL-18 (Figure 4B). Representative flow cytometry plots are shown in Figure 4C. 

### 3.6. Anti-Tumor Activity of γδ T Cells is Remarkably Enhanced by the Combination of IL-2/IL-12/IL-18 even in the Absence of a TCR Stimulus

γδ T cells exhibit tumor lysis against various types of tumors [38]. After stimulation with the above-mentioned cytokines, γδ T cells from three donors were co-cultured with 3 different tumor cell lines, that grow as a monolayer: A673: derived from a Ewing sarcoma, RH30: a rhabdomyosarcoma and SH-SY5Y: a neuroblastoma. We chose these tumor cell lines to see whether cytokine-activated γδ T cells may possibly have the potential for a new and alternative immunotherapeutic strategy in these cancer entities, which are aggressively growing malignancies with very poor prognosis in children and adolescents. By using the electric cell-substrate impedance sensing device, cytotoxic effects can be monitored since when stimulated γδ T cells lyse tumor cells, the dying tumor cells detach from the plate, resulting in reduced impedance. Different E:T ratios (5:1, 2.5:1, 1:1, 0.5:1) were used and the impedances were analyzed at 24 h intervals.

Anti-tumor activity against these three tumor cell lines was highly dependent on the E:T ratios. As for A673, independently of the presence of a former TCR stimulus, at a high E:T ratio of 5:1, all cytokine treatment resulted in massive tumor lytic activity in γδ T cells. Interestingly, at the low E:T ratio of 0.5:1, only IL-2/IL-12/IL-18-pretreated γδ T cells exhibited a sustained anti-tumor effect, in fact irrespective of a simultaneous TCR stimulus (Figure 5).

Regarding RH30, at a high E:T ratio of 5:1, a significant reduction in tumor growth by γδ T cells was seen after all cytokine treatments with TCR stimulus with the combination of IL-2/IL-12/IL-18 leading to the highest reduction of impedance. Without TCR stimulus, only the cells exposed to combination treatment of IL-2, IL-12 and IL-18 had an effective suppressive effect on tumor growth even at a low E:T ratio, i.e., of 0.5:1. On the other hand, the other cytokine treatments without TCR stimulus showed no tumor lytic effect even at a high E:T ratio of 5:1 (Figure 5).

With regard to SH-SY5Y, TCR-stimulated γδ T cells showed no significant tumor growth inhibition even at a high E:T ratio of 5:1 except when combined with IL-2/IL-12/IL-18. In the absence of a TCR stimulus, only IL-2/IL-12/IL-18- treated γδ T cells exerted a clear and lasting anti-tumor effect, even at a low E:T ratio, i.e., of 0.5:1 (Figure 5).

### 3.7. Combined IL-2/IL-12/IL-18 Induces Senescence in Tumor Cells Independent of a TCR Stimulus

Subsequently, we investigated whether IL-2/IL-12/IL-18-pre-exposed γδ T cells in the absence of a TCR signal can drive cancer cells into senescence and terminal growth arrest, over time resulting in reduced impedances compared to control. To this end we co-cultured γδ T cells (untouched or untouched isolated and IL-2/IL-12/IL-18-stimulated, or TCR/IL-2/IL-12/IL-18-stimulated) with T24 (bladder carcinoma), MCF7 (breast cancer) or WM115 (melanoma) cancer cells for 96 h and then evaluated the cell cycle status of the tumor cells by EdU-incorporation assay. Additionally we stained senescent cells with SA-*β*-galactosidase and cell cycle regulators expressed by the tumor cells were analyzed by qPCR. MCF7 and Wm115 were included since breast cancer and melanoma constitute a major share of adult malignancies. T24 was chosen as in bladder cancer immunotherapeutic strategies have already shown significant effects, i.e., through the instillation of BCG into the bladder of the patient, leading to the activation of γδ T cells [39]. In addition, these cell lines were less susceptible to activated γδ T cells than A673 and RH30, rendering sufficient numbers of remaining cells for further analysis by flow cytometry and qPCR. Since the combination of T_H_1 cytokines IFN-γ and TNF-α induces senescence [40], we used these cytokines (50 ng/mL of IFN-γ and 5 ng/mL of TNF-α) as a positive control for senescence induction. IL-2/IL-12/IL-18-stimulated γδ T cells terminally arrested T24, MCF7 or Wm115 cells in G1/G0 phase and massively reduced S-phase cells (Figure 6 and Figure 7A,B), a feature characterizing senescence. Molecular analysis revealed in MCF7 and in T24 cells the significant upregulation of cell cycle regulator *p21* (CDK inhibitor 1), an inhibitor of cyclin-dependent kinases essential for the control of the cell cycle in mammals, which amongst other functions, suppresses cancer growth when co-incubated with IL-2/IL-12/IL-18-stimulated γδ T cells, regardless of activation via IMMU510 on effector cells. Degree of upregulation in *p21* was comparable to the positive control, i.e., pure T_H_1 effector cytokines IFN-γ/TNF-α (Figure 8), indicating the potent release of these effector cytokines by IL-2/IL-12/IL-18 stimulated γδ T cells. The expression of a cell cycle inhibitor *p21* in Wm115 cells was slightly increased by co-culture with IL-2/IL-12/IL-18-stimulated γδ T cells but the difference did not reach statistical significance. In MCF breast cancer cells the expression of a cell cycle regulator *p53* was significantly elevated after combined treatment with IFN-γ/TNF-α, but not significantly up-regulated by co-culture with IL-2/IL-12/IL-18-stimulated γδ T cells. Statistically significant up-regulation of tumor suppressor genes *p53* or *p16* was observed in none of the co-culture experiments with IL-2/IL-12/IL-18-stimulated γδ T cells in all tumor cell lines tested. These results suggest that IL-2/IL-12/IL-18-stimulated γδ T cells induce senescence in MCF7 cells and T24 cancer cells through the *p21* cascade, whereas in WM115 the senescence pathway remains to be elucidated. An increase of SA-*β*-galactosidase positive, i.e., senescent cells which are morphologically large, flat, and multinucleated, was confirmed as shown in Figure 9.

### 3.8. Direct Comparison of Two Different Anti-γδTCR Monoclonal Antibodies Regarding Cytokine Production and Efficacy in Senescence Induction

As for the potential difference provoked by different anti- γδ TCR monoclonal antibodies, we directly compared clone IMMU510 and clone B1 in terms of cytokine production and cell cycle arrest induction in tumor cells by stimulated γδ T cells. As shown in Appendix A, the frequencies of IFN-γ^+^, IL-17A^+^, and TNF-α^+^ cells did not differ between IMMU510 and B1. In addition, cell cycle status of Wm115 cells after co-culture with γδ T cells was almost identical between IMMU510 and B1 (Appendix A). This suggests that these two monoclonal antibodies exhibit similar effect on γδ T cells.

### 3.9. Senescence Induction in Tumor Cells by γδ T Cells Treated with IL-2/IL-12/IL-18 is Mediated by IFN-γ/TNF-α

To reveal whether the senescence induction in tumor cells is dependent on IFN-γ/TNF-α produced by IL-2/IL-12/IL-18-stimulated γδ T cells, we neutralized IFN-γ/TNF-α by anti-IFN-γ antibody or anti-TNF-α antibody or both antibodies. As consistent with Figure 6 and Figure 7, IL-2/IL-12/IL-18-stimulated γδ T cells terminally arrested T24, MCF7 or Wm115 cells in G1/G0 phase or arrested MCF7 also in G2/M phase and blocked entry in S phase reflected by a massively reduced number of cells in S-phase (Figure 10A,B). Neutralizing IFN-γ abrogated this effect in T24 and MCF7. Neutralizing TNF-α abolished this effect in T24 and partially cancelled it in MCF7. Neutralizing both IFN-γ and TNF-α completely cancelled the cell cycle arrest effect in all of the cell lines, i.e., Wm115, T24, and MCF7. These findings indicate that IL-2/IL-12/IL-18-stimulated γδ T cells induce senescence in tumor cells through their production and release of IFN-γ/TNF-α.

## 4. Discussion

γδ T cells infiltrate a variety of cancers and modulate anti-tumor immunity [41,42,43,44], and a recent extensive gene expression study identified tumor-infiltrating γδ T cells as the most favorable prognostic marker in many types of cancer [45]. Along with their unique beneficial features such as the production of abundant pro-inflammatory cytokines, including IFN-γ and TNF-α, MHC-independent recognition of antigens, direct lysis of tumor cells, antibody-dependent cellular cytotoxicity (ADCC), and antigen presentation capacity [46], γδ T cells are proposed to constitute a promising target in cancer immunotherapy. However, adoptive transfer of pharmacologically manipulated γδ T cells did not hold promise. Driven by the rational to avoid supraphysiologic TCR triggers and emulating a finding by Silva-Santos et al. [47] that the differentiation of human γδ T cells into cytotoxic type I effector T cells is exclusively dependent on cytokines and not involving TCR activation, and moreover occurs in the periphery, we systematically analyzed the impact of IL-2, IL-12, and IL-18 alone and combined, in the presence and absence of TCR stimulation. The results from this comparative study demonstrate that the combined stimulation of γδ T cells via IL-2/IL-12/IL-18 is the most potent stimulus to enhance anti-tumor activity, proliferation and IFN-γ production in γδ T cells in the absence of TCR ligation.

Corresponding with our findings, a previous study [20] reported that proliferation, IFN-γ production, granzyme B and NKG2D expression are induced in response to combined treatment with IL-12/IL-18 in ex vivo-expanded Vγ9Vδ2 T cells and IFN-γ production in freshly isolated Vγ9Vδ2 T cells. Conflicting with our study, in this study combined treatment with IL-12 and IL-18 failed to induce proliferation within a short 16 h period and the expression levels of T-bet were increased in response to IL-12 or IL-12/IL-18 treatment. Such discrepancy may result from methodological differences. While in this study γδ T cells were stimulated with cytokines after ex vivo-expansion with zoledronate and subsequent cryopreservation and thawing, in our present study exclusively freshly isolated untouched γδ T cells were stimulated with cytokines and TCRγδ antibody. In addition, the untouched γδ T cells were incubated with cytokines for three days instead of a short 16-h period in the previous study. 

Efficient tumor-lytic activity after pharmacologic Vδ2 TCR-stimulation correlates with granzyme B and IFN-γ/TNF-α production in γδ T cells, parameters which we and others found to be enhanced when IL-2, IL-12, and IL-18 are added [16,17,18,19]. Amazingly, IL-2/IL-12/IL-18-exposed γδ T cells produce IFN-γ, and express cytotoxins and FasL, resulting in potent anti-tumoral activity comparable to that of TCR-stimulated counterparts, as shown for different E:T ratios during a 96-h monitoring period towards three tested tumor cell lines (Figure 5); Anti-tumoral activity of bypass stimulated γδ T cells comes along with insignificantly changed NKG2D expression, thus excluding an enhanced anti-tumor activity of TCR-bypass stimulated γδ T cells via this receptor. The comprehensive examination of the tumor cells remaining after killing assay—96 h or 120 h after impedance assay respectively—revealed that besides killing IL-2/IL-12/IL-18 alone is able to instruct γδ T cells to drive cancer cells also into senescence and terminal growth arrest, as recently shown by us as a consequence and combinational effect of IFN-γ and TNF-α secreted by T_H_1 immune effector cells [40]. Interestingly, cancer control via cell cycle arrest is in perplexing harmony with Robert Gatenby’s provocative advice in “A change of strategy in the war on cancer’’ [48], where he states that controlling a tumor’s growth may be more efficient in the long term, rather than trying to eradicate it. Senescence-inducing γδ T cells mirror Gantenby’s strategic perception accurately: cell cycle-arrested tumor cells are unable to divide but since they are not dying but metabolically highly active [49,50,51,52], compete for nutrients in the same niche with residual tumor cells [48]. In synopsis of these findings it may be allowed to speculate that “killing” may not be the ultimate goal for γδ T cells but may rather represent the extreme on a scale that—by integrating as yet to be defined diverse (microenvironmental) signals—comprises multiple layers of adjustable function.

TCR-independent, cytokine-instructed γδ T cell function clearly differs from adaptive T cell responses but fits well with γδ T cells sentinel function in the tissues. Arresting the cancer cells paves the way for both arms of the immune system; NK cells in a permanent “ready to kill” mode—wipe out any cell that cannot bind their inhibitory receptors by lack of MHC class I, while engulfment of senescent tumor cells by macrophages facilitates the establishment of adaptive immunity and long-lasting memory. This underpins the macrophages’ key role in the tumor microenvironment as the source of IL-12 and IL-18, thus forming the crystallization point for innate and adaptive immunity. 

In summary, the present study investigates the modulation of γδ T cell effector functions solely by myeloid cytokines, IL-12, IL-18, or the combination of IL-2, IL-12, and IL-18 in the presence and absence of a pharmacological TCR stimulus. This comparative study demonstrates for the first time that TCR bypass stimulation i.e., γδ T cells stimulated solely with combined IL-2/IL-12/IL-18 is enough to induce γδ T cells that exhibit the same efficiency in anti-tumor activity as their counterparts after TCR crosslinking. Moreover, irreversible growth arrest in cancer cells suggests long-lasting tumor control. With regard to cell based adoptive therapy this study shows that in the course of three days the combination of IL-2/IL-12/IL-18 leads to 6-fold expansion of γδ T cells compared to control. These findings open an avenue for the generation of increased numbers of γδ T cells and approve/recommend the adoptive transfer of IL-2/IL-12/IL-18-stimulated γδ T cells for an unbiased clear boost of anti-tumor activity of γδ T cells in clinical settings. Protocols for the generation and transfer of cytokine-activated Vδ1^+^ γδ T cells, so called DOT cells, have been established by Silva-Santos’s group and have already been proven to be safe [53]. Analogously patient-derived γδ T cells [54] or donor-derived γδ T cells for hematopoietic stem cell transplantation [55] can be isolated according to GMP guidelines by “negative selection”, i.e., αβ T cell depletion from peripheral blood or mobilized donors’ apheresis. Such γδ T cells could be ex vivo stimulated with IL-2, IL-12, and IL-18, thereby omitting the unwanted side effects of the systemic administration of cytokines. IL-2 (Proleukine) is already in clinical use, and IL-12 and IL-18 are available in GMP quality (IL-12: R&D Systems and Miltenyi Biotec, and IL-18: AkronBiotech). These conditions make the adoptive transfer of a cytokine-activated γδ T cell compartment feasible and highly promising for clinical use.

## 5. Conclusions

It has been known for some time that the TCR is involved in the acquisition of suppressive function of γδ T cells [12,13,14] and we showed recently that suppressive activity of Vδ2^+^ γδ T cells on αβ T cells is licensed by TCR signaling and correlates with signal strength [56]. Thus TCR-independent activation of γδ T cells is highly desirable for unbiased immunotherapy in cancer to prevent unwanted effects on adaptive T cell responses. To clarify the modulation of γδ T cells’ effector functions solely by cytokines, we systematically analyzed the impact of IL-2, IL-12, and IL-18 in parallel with TCR stimulation on proliferation, cytokine production, and anti-tumor activity of γδ T cells and demonstrate that IL-12 and IL-18 when combined, so-called bypass stimulation, constitute the most potent stimulus to enhance anti-tumor activity and induce proliferation and IFN-γ production by γδ T cells in the absence of TCR signaling. 

Since stimulation with IL-12 and IL-18 without simultaneous TCR stimulus induces potent anti-tumoral activity in γδ T cells comparable to that after TCR crosslinking i.e., efficient killing of tumor cells and driving cancer cells into senescence -, the use of IL-12/IL-18-stimulated γδ T cells is highly recommended for adoptive cell therapy to boost anti-tumor activity by γδ T cells. 

## Figures and Tables

**Figure 1 cancers-12-00130-f001:**
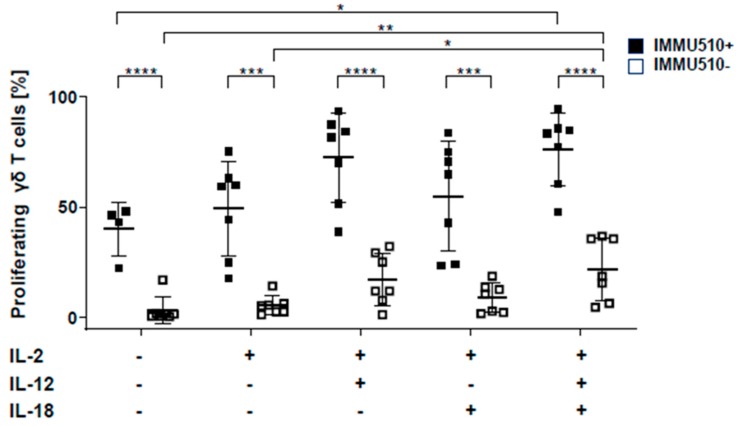
The combination of IL-12 and IL-18 induces the proliferation of γδ T cells. CTV-labelled γδ T cells were cultured for 4 days with culture medium alone (no cytokines), IL-2 (50 U/mL), IL-12 (10 ng/mL), IL-18 (10 ng/mL), or IL-12 with IL-18 (each 10 ng/mL, respectively) in the presence or absence of anti-TCRγδ monoclonal antibody IMMU510. CTV^low^ cells were calculated as proliferating cells. The data were obtained from 7 different donors. One-way ANOVA followed by Tukey’s multiple comparison test was used for identification of significances. Bars represent the mean ± SD. * *p* < 0.05, ** *p* < 0.01, *** *p* < 0.001, **** *p* < 0.0001.

**Figure 2 cancers-12-00130-f002:**
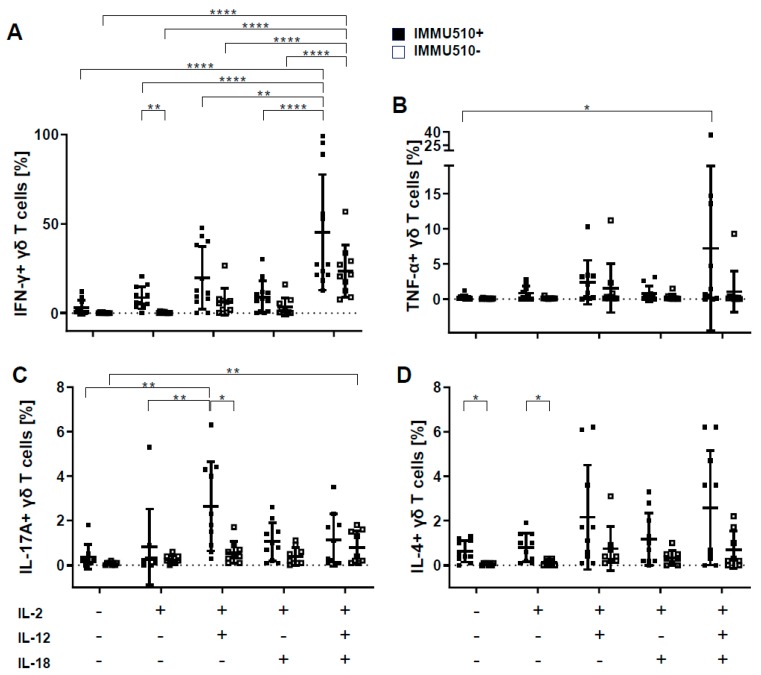
Cytokines produced by γδ T cells in response to cytokines and or TCR stimulation. γδ T cells were cultured as described in Material and Method section and Figure legend 1. γδ T cells were incubated with Brefeldin A 1 h before intracellular expression of (**A**) IFN-γ, (**B**) TNF-α, (**C**) IL-17and (**D**) IL-4, was analyzed. (**E**) Representative plots of IFN-γ/IL-17A and IFN-γ/IL-4 produced by γδ T cells stimulated with IL-2/IL-12/IL-18 in the presence and absence of IMMU510 are shown. (**F**) Representative plots of IFN-γ/TNF-α produced by γδ T cells are shown. Medium alone (no stimulation) served as control for IL-2/IL-12/IL-18 stimulation, TCR-stimulation, and IL-2/IL-12/IL-18/TCR-stimulation. One-way ANOVA followed by Tukey’s multiple comparison test was used for identification of significances. Bars represent the mean ± SD. * *p* < 0.05, ** *p* < 0.01, **** *p* < 0.0001.

**Figure 3 cancers-12-00130-f003:**
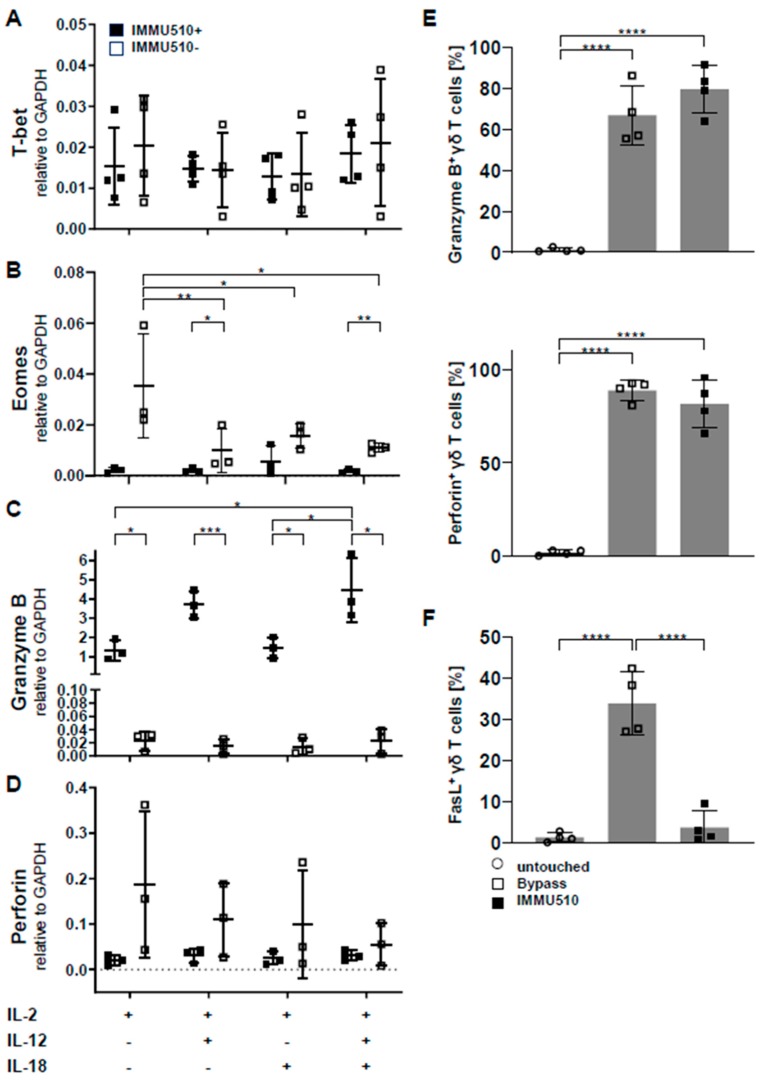
T-bet/Eomes expression of Vδ2 γδ T cells in the presence or absence of cytokines and or TCR stimulation. While granzyme B is *not* expressed in γδ T cells in the absence of TCR stimulation irrespective of the presence of cytokines, it is increased by IL-2/IL-12 or IL-2/IL-12/IL-18 in the presence of TCR stimulation. γδ T cells were cultured as described in Material and Method section and Figure legend 1. The expression of (**A**) T-bet, (**B**) Eomes, (**C**) granzyme B and (**D**) perforin was analyzed by qPCR. The expression of (**E**) granzyme B, perforin, and (**F**) FasL in negatively isolated fresh, IL-2/IL-12/IL-18-stimulated, and IMMU510-stimulated γδ T cells. The data were obtained from 4 different healthy donors. Because of limited RNA sample, the data were obtained from 3 different healthy donors in (**B**–**D**). One-way ANOVA followed by Tukey’s multiple comparison test was used. Bars represent the mean ± SD. * *p* < 0.05, ** *p* < 0.01, *** *p* < 0.001, **** *p* < 0.0001.

**Figure 4 cancers-12-00130-f004:**
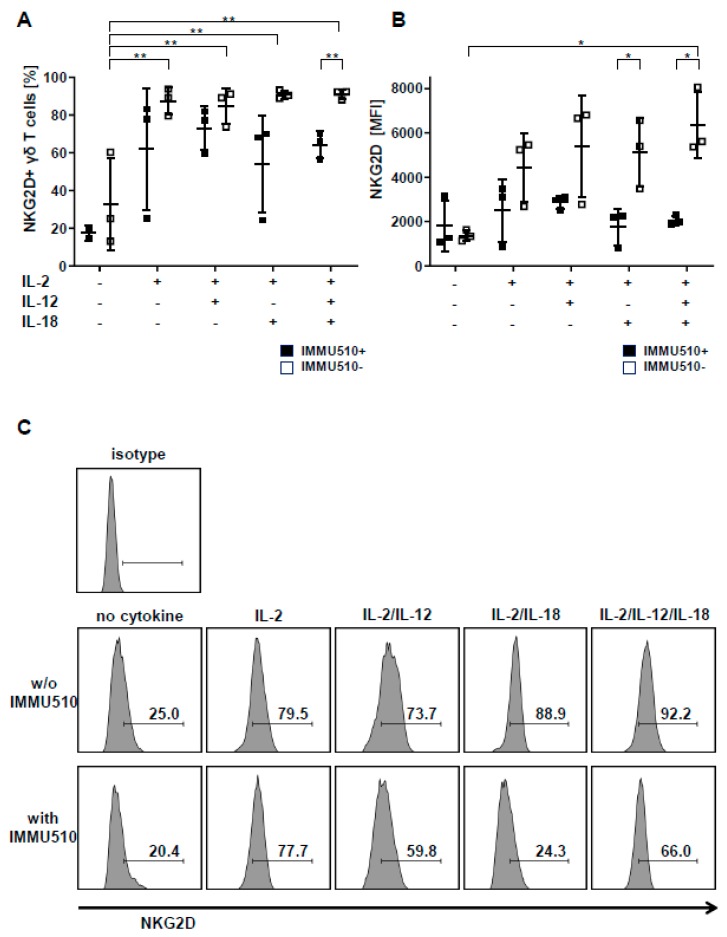
TCR stimulation represses upregulation of NKG2D on γδ T cells. γδ T cells were cultured as described in Material and Method section and Figure legend 1 and expression of NKG2D analyzed by FACS. (**A**) NKG2D-positive γδ T cells after exposure to diverse cytokines, (**B**) expression height of NKG2D given as MFI signals of γδ T cells. Data were obtained with cells from 3 healthy donors. One-way ANOVA followed by Tukey’s multiple comparison test was used. Bars represent the mean ± SD. * *p* < 0.05, ** *p* < 0.01. (**C**) Representative plots of NKG2D expressed by γδ T cells are shown together with isotype control.

**Figure 5 cancers-12-00130-f005:**
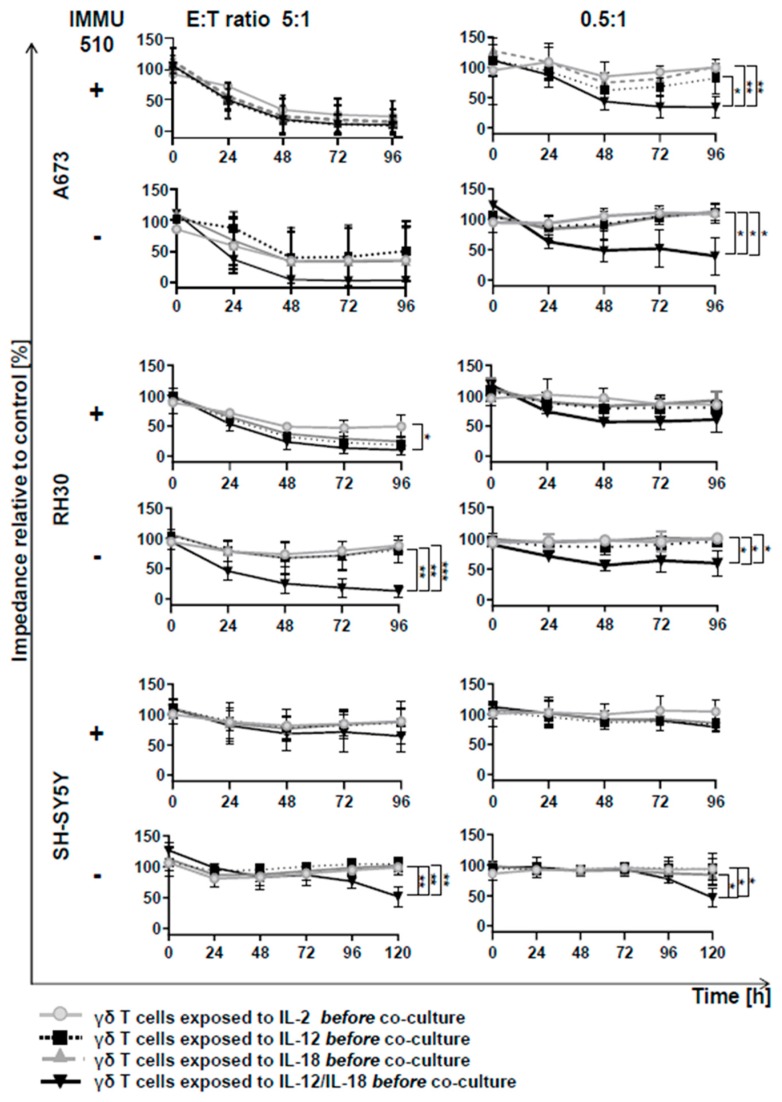
Potent anti-tumor activity of pan-γδ T cells after IL-2/IL-12/IL-18 exposure occurs independently of a concomitant TCR stimulus. γδ T cells after preemptive exposure to respective cytokines in the presence or absence of TCR stimulation as described above were co-cultured with A673, RH30 and SH-SY5Y cells (grown to 80% confluency) at different effector:tumor cell ratios (5:1, 2.5:1, 1:1, 0.5:1). By using the Electric Cell-substrate Impedance Sensing device, cytotoxic effect was monitored at 24 h intervals for 96 or 120 h respectively. Given are the results for highest and lowest E:T ratios. Anti-tumor activity of γδ T cells from 3 healthy individuals was analyzed. One-way ANOVA followed by Tukey’s multiple comparison test was used. Bars represent the mean ± SD. * *p* < 0.05, ** *p* < 0.01, *** *p* < 0.001.

**Figure 6 cancers-12-00130-f006:**
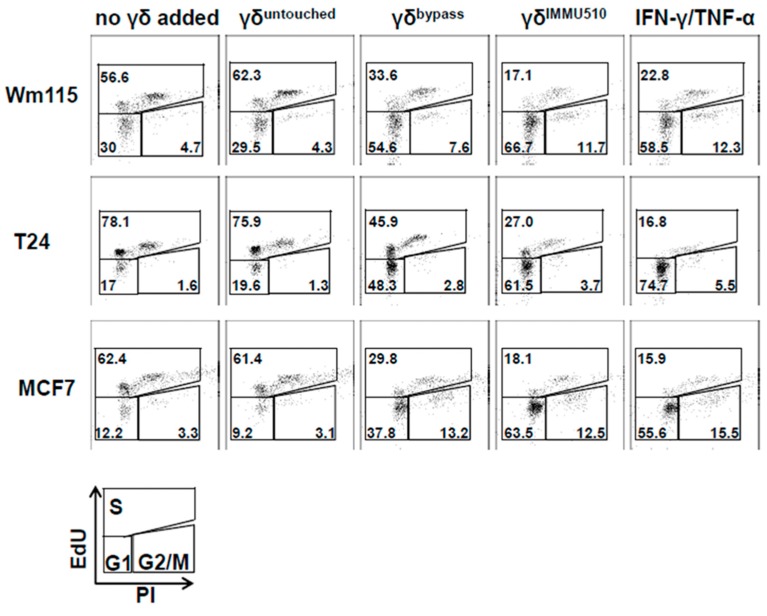
IL-2/IL-12/IL-18 combined but not alone elicit γδ T cells that induce cell cycle arrest in tumor cells as shown in FACS analysis. γδ T cells (untouched, IL-2/IL-12/IL-18-stimulated, TCR stimulated or TCR/IL-2/IL-12/IL-18-stimulated) were co-cultured with T24 (bladder carcinoma), MCF7 (breast cancer) or WM115 (melanoma) cells for 96 h and the cancer cells then analyzed for their cell cycle profile. Representative EdU-incorporation assay FACS plots are shown for each tumor cell line and each stimulation setting.

**Figure 7 cancers-12-00130-f007:**
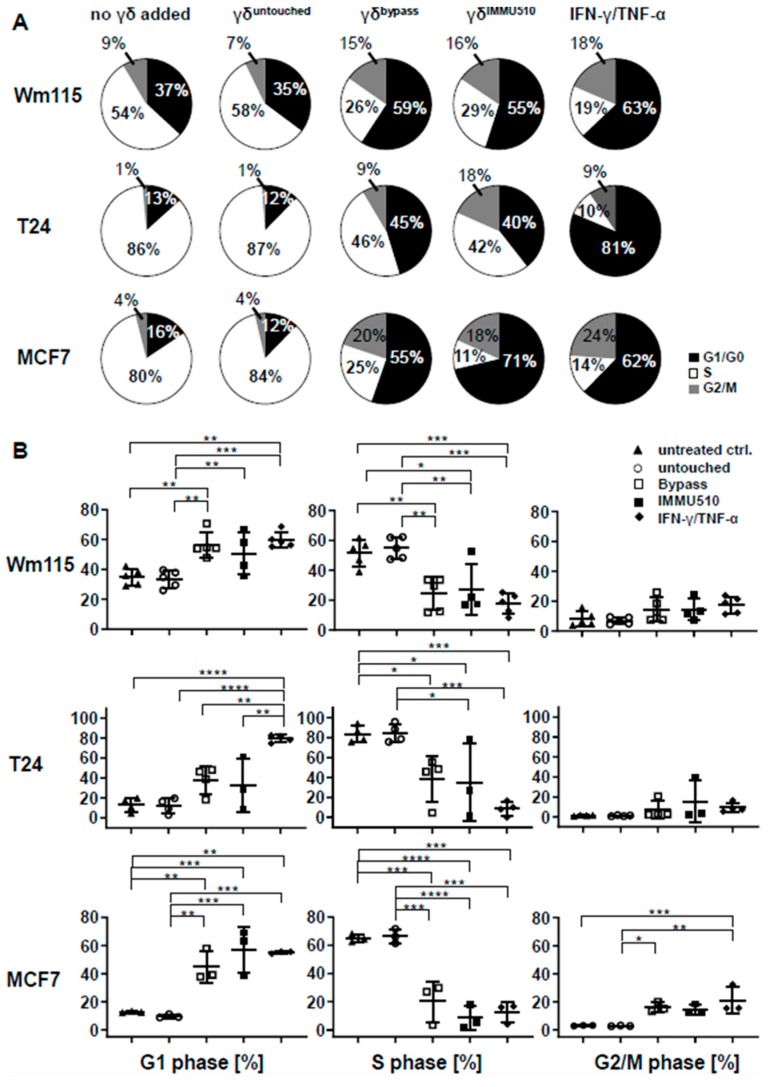
IL-2/IL-12/IL-18 combined but not alone elicit γδ T cells that induce cell cycle arrest in tumor cells. A summary of integrated data from multiple experiments is shown (WM115 *n* = 5, T24 *n* = 4, MCF7 *n* = 3) γδ T cells (untouched, IL-2/IL-12/IL-18-stimulated, TCR stimulated or TCR/IL-2/IL-12/IL-18-stimulated) were co-cultured with T24 (bladder carcinoma), MCF7 (breast cancer) or WM115 (melanoma) cells for 96 h and the cancer cells then analyzed for their cell cycle profile. (**A**) The average frequencies of individual cell cycle phases analyzed by EdU-incorporation assay are shown for each cell line. Each pie blot represents the data obtained from 5 experiments using γδ T cells of 5 different donors. (**B**) Numerical values of cell cycle stages of tumor cells analyzed by EdU-incorporation assay. One-way ANOVA followed by Tukey’s multiple comparison test was used for statistical analysis. Bars represent the mean ± SD. * *p* < 0.05, ** *p* < 0.01, *** *p* < 0.001, **** *p* < 0.0001.

**Figure 8 cancers-12-00130-f008:**
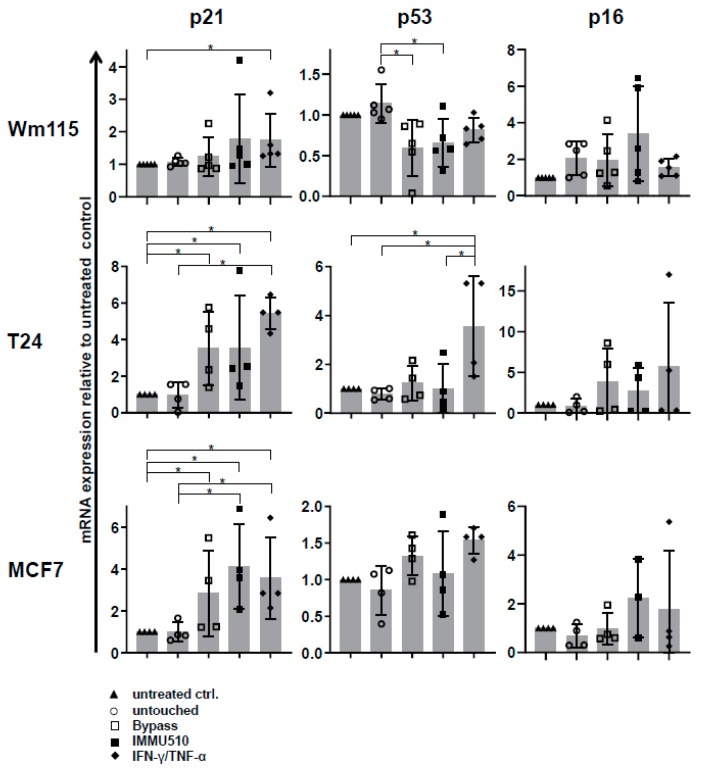
IL-2/IL-12/IL-18-stimulated γδ T cells significantly upregulate cell cycle inhibitor p21 in tumor cells. The expression of *p21*, *p53*, and *p16* in tumor cells was analyzed by qPCR. The data were obtained from 5 different healthy donors for Wm115, and 4 donors each for T24 and MCF7. One-way ANOVA followed by Tukey’s multiple comparison test was used for statistical analysis. Bars represent the mean ± SD. * *p* < 0.05.

**Figure 9 cancers-12-00130-f009:**
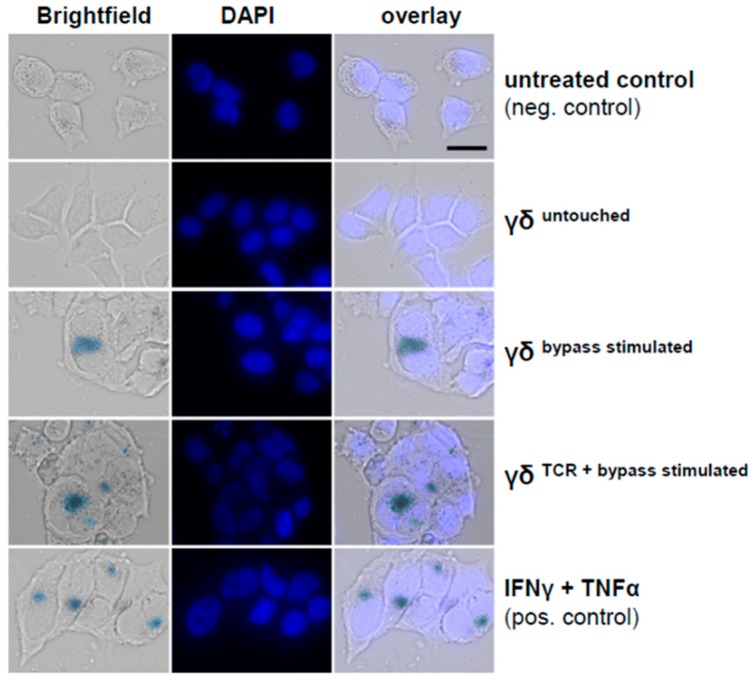
MCF7 cells express the senescence marker SA-*β*-galactosidase after co-culture with IL-2/IL12/IL-18 stimulated γδ T cells. Freshly isolated untouched, TCR-bypass-stimulated, or IMMU510/IL-2/IL-12/IL-18-stimulated γδ T cells were co-cultured with MCF-7 breat cancer cells in a 1:1 ratio. Medium served as a negative control, medium supplemented with IFN-γ/TNF-α as a positive control for senescence induction as reported previously [40]. After a 4-days-incubation, cancer cells were analyzed for the expression of senescence associated marker SA-*β*-galactosidase. The experiment was repeated 3 times using 3 different donors. Shown are data from one donor representative for all three experiments.

**Figure 10 cancers-12-00130-f010:**
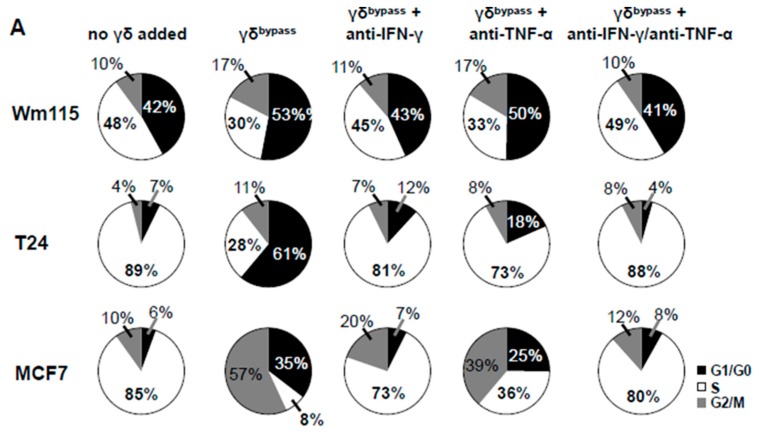
Cell cycle arrest in tumor cells induced by IL-2/IL-12/IL-18-stimulated γδ T cells is mediated by IFN-γ and TNF-α. IL-2/IL-12/IL-18-stimulated γδ T cells were co-cultured with T24, MCF7 or WM115 cells for 96 h in the presence of anti-IFN-γ antibody or anti-TNF-α antibody or both antibodies and the cancer cells then analyzed for their cell cycle profile. (**A**) The average frequencies of individual cell cycle phases analyzed by EdU-incorporation assay are shown for each cell line. Each pie blot represents the data obtained from 2 experiments using γδ T cells of 2 different donors. (**B**) Numerical values of cell cycle stages of tumor cells analyzed by EdU-incorporation assay. One-way ANOVA followed by Tukey’s multiple comparison test was used for statistical analysis. Bars represent the mean ± SD. * *p* < 0.05, ** *p* < 0.01.

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
