# Peer review of "In the Absence of a TCR Signal IL-2/IL-12/18-Stimulated γδ T Cells Demonstrate Potent Anti-Tumoral Function Through Direct Killing and Senescence Induction in Cancer Cells"

_cancers, 2020, doi:10.3390/cancers12010130_

Round 1
Reviewer 1 Report
Schilbach et al. demonstrate in this paper that gd T cells stimulated with IL-12 or IL-18, or both can acquire anti-tumor potential by producing cytokines, such as IFNg and TNFa, and granzyme and perforin that can mediate killing of tumor cells. The conclusion of this paper is interesting; however, there are a number of of points to be addressed by the authors.
In the experiments shown in Figures 1 to 4, although the authors mentioned gd T cells were stimulated with IL-12 and IL-18, always IL-2 was added together with these two cytokines. I believe that the authors should better correct or edit the text and figure legends so that readers would not misunderstand the experiments the authors have done. Related to this issue, in the experiments shown in Figures 5-9, the authors should clearly describe whether IL-2 was added.
The authors purified gd T cells from human PBMCs using MACS beads and used the cells in the experiments. However, at multiple places in the text, the authors mentioned as if they used Vd2+ gd T cells in the experiments. Without any experimental data or scientific justification, this misstatement is not allowed.
Throughout the paper, the authors did not mention how many times they repeated the experiments and whether the presented results in the figures are representative.
For Figure 4, if the authors show representative FACS plots, the figure would be more informative.
In Figures 5 and 6, a different set of tumor cell lines were used. The authors need to explain why they did so.
The authors mentioned that tumor cell senescence shown in Figures 6-9 was induced by IFNg and TNFa.If the authors would like to draw such a conclusion, the authors should show that cell senescence of tumor cell lines is not induced in co-culture with stimulated gd T cells in the presence of neutralizing antibodies for IFNg and TNFa.
Minor point: The authors should show the same number of the total events in each FACS plot, at least in the same figure.
Reviewer 2 Report
Authors demonstrated stimulation with IL-12 and IL-18 induces effective anti-tumor activity in gdT-cells even in the absence of TCR stimulation. Their data are informative and convincing but authors should provide some other data or discussions about their data set for publishing the manuscript.
Authors demonstrate multi cytokine productions (including IFN-g, TNF-a, IL-17A, and IL-4) in gdT-cells in figures 2. IL-12/18 plus IMMU510 stimulated gdT-cells seems to be the most IFN-g-producing cells. It’s important if the stimulated gdT-cells have multifunction (potential to produce several cytokines), because type II response is not desirable for anti-tumor immunity. Are the IFN-g-producing gdT-cells also produce IL-17A or IL-4? Or does other population produce these cytokines in the IL-12/18 plus IMMU510 stimulated gdT-cells? Please show the flow cytometry plots like figure 2 E.
Authors show in figure 3A that the IFN-g production of gdT-cells does not depend on T-bet and expected that Eomes or IkB would be critical for it. Please assess the expression levels of at least eomes in qPCR as they demonstrated those of T-bet.
Authors show in figure 6 and 7 that IL-12/18-stimulated gdT-cells induce cell cycle arrest in tumor cells. Does the mechanism depend on IFN-g and TNF-a, produced from the IL-12/18-stimulated gdT-cells? Or does other mechanism exist for that? Please show the data of blocking assay using specific antibodies for the signaling involved in IFN-g and TNF-a stimulation.
Minor comment
Authors mentioned “Th1 gdT cells” in discussion. But the Th1 means type I helper T cells which have activity of IFN-gproduction and cell killing. So, that is not inappropriate.
Reviewer 3 Report
The paper by Schilbach, K. is interesting and it will really open up a new dimension of the therapeutic perspective. I have a few comments to improve the manuscript:
I see that whenever the IMMU510 is used despite the proliferation experiment in Fig 1 or qPCR in Fig 3, the variance between the patients is quite high. Is there any way to reduce the variance between the patients? Is it possible to experiment with a higher number of patients? Is it possible to use some other anti-TCRgd monoclonal antibody? The FACS plot from Fig 2E is also not quite impressive especially the IMMU510+IL-12/IL18 plot. This is an interesting study where the research design is quite appropriate but one has to be careful with the representation of the figures. It would also increase the readability and visibility if the author expands the discussion a bit more and explain what they want to do in their future work.
Round 2
Reviewer 1 Report
The authors nicely revised the paper. I believe this version is good enough to be published.
Reviewer 2 Report
Authors responded to my concerns and questions well.